# A Supply Chain-Oriented Model to Predict Crude Oil Import Prices in South Korea Based on the Hybrid Approach

**Jisung Jo [1], Umji Kim [2], Eonkyung Lee [1], Juhyang Lee [3] and Sewon Kim [3,*]**

1   Logistics and Maritime Industry Research Department, Korea Maritime Institute, Busan 49111, Republic of Korea; jisungjo@kmi.re.kr (J.J.); eklee@kmi.re.kr (E.L.)
2   Northern and Polar Regions Research Division, Korea Maritime Institute, Busan 49111, Republic of Korea; umjikim@kmi.re.kr
3   Department of Intelligent Mechatronics Engineering, Sejong University, Seoul 05006, Republic of Korea; wngid0822@naver.com
*   Correspondence: sewonkim@sejong.ac.kr

**Abstract:** Although numerous studies have explored key variables for forecasting crude oil prices, the role of supply chain factors has often been overlooked. In the face of global risks such as COVID-19, the Russia–Ukraine war, and the U.S.–China trade dispute, supply chain management (SCM) has evolved beyond an individual company's concern. This research investigates the impact of a supply chain-oriented variable on the forecasting of crude oil import prices in South Korea. Our findings reveal that models incorporating the Global Supply Chain Pressure Index (GSCPI) outperform those without it, emphasizing the importance of monitoring supply chain-related variables for stabilizing domestic prices for policy makers. Additionally, we propose a novel hybrid factor-based approach that integrates time series and machine learning models to enhance the prediction performance of oil prices. This endeavor is poised to serve as a foundational step toward developing methodologically sound forecasting models for oil prices, offering valuable insights for policymakers.

**Keywords:** supply chain oriented; GSCPI; machine learning; forecasting; crude oil import price

## 1. Introduction

Until now, supply chain management (SCM) has primarily been perceived as an issue confined to individual companies. Both large and small disruptions occur routinely at the enterprise level, often being naturally resolved through price adjustments. However, the landscape of supply chain disruptions has undergone a significant transformation, reaching a global scale due to risks such as the obstruction of the Suez Canal, the COVID-19 pandemic, the Russia–Ukraine war, and the U.S.–China trade dispute. This escalation of supply chain disruptions has surpassed the capabilities and scope of individual companies. It is no longer sufficient to rely solely on price adjustments for resolution. Notably, the direct correlation between supply chain risk management for critical goods or strategic materials—integral for national security—and the maintenance of industrial competitiveness, social stability, and diplomatic and security leverage has elevated the issue. Consequently, developing robust response systems and building resilience to supply chain shocks have become imperative tasks, requiring coordinated efforts at both corporate and national levels.

South Korea has been significantly affected by recent global supply chain risks. In particular, it is important for the economy of South Korea to secure stable prices and the supply of raw materials such as crude oil, which is 100% import-dependent. There are many studies that have analyzed the correlation between the Korean economy and international oil prices. Shin [1] derived, through quantitative analysis, that the impact of international oil prices on the Korean economy continues to grow. Lee [2] analyzed how

Korea's GDP and producer price index change in response to oil price hikes and petroleum product prices.

Also, as oil prices have a great impact not only on Korea but also on the world economy, numerous researchers have focused on predicting crude oil prices. Shin et al. [3] proposed a semi-supervised learning method devised for oil price prediction. Mahdian and Khamehchi [4] compared a modified neural network model with a pure neural network in predicting both daily and monthly crude oil prices, demonstrating its superior performance, particularly in situations with a small number of input data for training or great fluctuations in variables. Xiong et al. [5] indicated that the proposed EMD–SBM–FNN model using the MIMO strategy is the best in terms of prediction accuracy with accredited computational load.

In this research, our objective is to examine whether supply chain-related variables can enhance the forecasting performance of crude oil prices. Furthermore, we propose a novel hybrid factor-based approach to enhance the accuracy of forecasting crude oil prices. Our research contributes to the analysis and forecast of the crude oil price by using the relationship with global supply chain parameters. The crude oil price variation is a crucial parameter that affects the socio-economy. The instability of the crude oil price causes unstable situations such as global energy supply instability and inflation. Therefore, our work aims to provide an analysis frame that predicts the crude oil price time-varying tendency regarding global supply chain pressure. Ultimately, this novel method aims to cope with this uncertain and unstable supply chain situation and enable sustainable supply chain management.

### 1.1. Types of Global Supply Chain Risk

Many previous studies have explored the types of global supply chain risks and the triggers that cause these supply chain disruptions. Yang et al. [6] classified internal risk types into logistical, financial, and information risks, and external risk types into policy, economy, culture, technology, natural disasters, and demand-related risks. Elsewhere, Harland et al. [7], Faisal et al. [8], and Manners-Bell et al. [9] also distinguished between internal and external risks. According to this, internal risks refer to cases that are directly related to operations, such as excessive inventory holding, product defects, and production volatility. External risks are those that can affect the supply chain from the outside, such as terrorism, war, piracy, or the global economic crisis. The WEF report in 2012 [10] also divided risks into internal and external risks: internal risks include credit rating, capital flow, intellectual property, asset value, and production quality, while external risks include natural disasters, disputes, brand reputational damage, and asset damage.

Zsidisin and Hendrick [11] classified risks into six areas: transportation, inventory, forecast, information, market, and suppliers. Dae-hyun et al. [12] added global risks to supply, operation, and demand risks. Global risks include innovative technologies, frequent legal and institutional changes, natural disasters, political risks, and strategic risks, as mentioned by Jeongwook [13]. Houlihan and Laurent [14] categorized supply chain risk factors into changes in short-term forecasts, changes in customer preferences, changes in technology, changes in government policies, changes in organizational frameworks, changes in organizational members, and changes in competitive strategies. Cooper and Ellram [15] classify risk factors as inaccurate fluctuations in customer demand, inaccurate supply lead times, partner financial conditions, inaccurate information, shortened product lifecycles, frequent market changes, globalization, intensified competition, and innovative technologies. These were divided into development risks and regulatory changes.

Christine et al. [7] defined four supply chain risk types: financial loss, material loss, psychological loss, and psychological loss. In addition, Tang and Musa [16] defined risk as any factor that disrupts or disrupts the supply chain process and grouped risk factors into material flow, financial flow, and information flow. Lin and Zhou [17], Seok-Mo and Choong-Bae [18], and Choong-Bae and Hee-Chan [19] classified the internal environment by supply chain nodes, such as supply and demand, while categorizing various risk factors,

such as natural disasters, terrorism, international politics, and war, into one risk factor called 'external environment.' In other words, these previous studies focused on risks that can occur within the supply chain while neglecting to classify risk types.

As a result of analyzing previous studies, risks can be largely divided into external and internal risks, with macro risk factors and micro risk factors. External risks refer to global factors that affect the supply chain, including natural disasters, war and terrorism, political instability, economic downturn, sovereignty risks, and regional instability. Internal risk refers to the risk that may occur in relation to all activities that a company conducts within its supply chain. Previous studies have identified risk factors by classifying them into different supply chain stages, such as demand, manufacturing, and supply. In the case of demand, risk factors such as inaccurate demand forecasts, rapid demand, short product life cycles, competitor movements, and market changes were derived. Risks at the manufacturing stage include technical knowledge, production capacity, product quality, demonstrations, and design changes. In this study, we focus particularly on risks in the logistics aspect that arise from external factors in the supply chain in the analysis.

### 1.2. Influential Factors and Models in Crude Oil Price Forecasting

Supply and demand factors have been widely recognized as significant indicators for oil price prediction. Hamilton [20] and Kilian [21] emphasized that oil supply and demand shocks are crucial determinants in explaining oil price shocks. Furthermore, Miao et al. [22] suggested a total of twenty-six determinants for forecasting models for the West Texas Intermediate (WTI) crude oil spot prices, grouping them into six categories: supply factors, demand factors, financial factors, commodity market factors, speculative factors, and political factors. With respect to supply and demand factors, they considered factors such as global production, global stock, global export, OPEC surplus, US stock, capacity utilization rate, Baltic Dirty, Kilian index; GDP growth in China, US, and EU; Steel World; global imports of China, US, and EU; and ISM. Despite the significant impact of supply chain disruptions on many economies, there is a scarcity of research papers that consider supply chain factors as determinants in oil price forecasting models. In this research, we aim to investigate whether supply chain-related variables could enhance forecasting performance.

A variety of models, including statistical and econometric models, artificial intelligence (AI) models, and hybrid models, have been employed to predict crude oil prices. Traditional time series econometrics models, such as autoregressive integrated moving average (ARIMA), generalized autoregressive conditional heteroscedasticity (GARCH), random walk (RW), a vector autoregression (VAR) model, and a vector error correction (VECM) model, are commonly used for oil price prediction. However, these models often face challenges in handling complexity and nonlinearity. As a response, AI models have been increasingly applied to the forecasting domain. Safari and Davallou [23] noted that time-series models might be insufficient to capture the nonlinear features of crude oil prices. To address this limitation, AI models are employed for oil price prediction. Azadeh et al. [24] introduced a flexible algorithm based on artificial neural networks (ANNs) and fuzzy regression (FR) to optimize long-term oil price forecasting in noisy, uncertain, and complex environments. Zhao et al. [25] utilized an advanced deep neural network model called stacked denoising autoencoders (SDAE) and an ensemble method named bootstrap aggregation (bagging) to demonstrate superior forecasting ability. Li et al. [26] proposed a novel crude oil price forecasting method based on online media text mining, aiming to capture more immediate market antecedents of price fluctuations.

In addition to individual approaches, there have been endeavors to explore hybrid methods. Safari and Davallou [23] identify three categories of hybrid methods, encompassing a combination of soft-computing techniques, a fusion of econometric models, and an amalgamation of soft-computing and econometric methods. They integrated the exponential smoothing model (ESM), the autoregressive integrated moving average model (ARIMA), and the nonlinear autoregressive (NAR) neural network to enhance the accuracy

of forecasting crude oil prices. Zhang et al. [27] introduced a hybrid method to predict crude oil prices, combining the ensemble empirical mode decomposition (EEMD) method, the least squares support vector machine together with the particle swarm optimization (LSSVM–PSO) method, and the generalized autoregressive conditional heteroskedasticity (GARCH) model. Abdollahi [28] constructed a hybrid model incorporating complete ensemble empirical mode decomposition, support vector machine, particle swarm optimization, and Markov-switching generalized autoregressive conditional heteroscedasticity to more effectively capture the nonlinearity and volatility of the time series. Despite numerous studies on developing hybrid models, a consensus on the best-fit model for forecasting oil prices has yet to be reached. In this research, we propose a novel hybrid factor-based approach to enhance the accuracy of forecasting crude oil prices. This will be achieved by comparing time series models and machine learning models based on the encompassing test.

This research aims to investigate whether supply chain-related variables have statistically significant effects on South Korea's crude oil import price. Additionally, we propose a novel hybrid factor-based approach to forecasting crude oil prices, incorporating supply chain aspects. This involves comparing the forecasting accuracy between traditional time series models and machine learning models. In the following section, we describe the time series models ARIMA, VAR, and VECM and the machine learning models KNN, SVM, and RF. The results are then discussed, followed by sections on discussion and conclusions.

The data are presented in Section 2.1. The main data sources are Petronet, KEEI, and Neworkfed. The time series data of the main crude oil price indicators are applied to analyze the effect due to global supply chain pressure. Supply and demand factors and supply chain factors are elaborated in Sections 2.1 and 2.2, respectively.

In Sections 2.2.1–2.2.5, the analysis of the models that analyze the effect of the global supply chain variables on the crude oil price is presented. Three time series models and three machine learning prediction methods are proposed. The target is to estimate the $\Delta$lnKprice by using the variation of the other parameters, such as supply demand and global supply chain pressure.

In Section 3, the experiment results are exhibited. The exogeneity test and Johansen's cointegration rank test are applied to the VAR and VECM models, which are applied to forecast the lnKprice based on the other indicators. The forecast performance of time series-based forecast methods and the machine learning methods are compared by applying them to the moving, expanding, and fixed window schemes. The limitations and the future works are elaborated in Sections 4 and 5. Figure 1 presents the schematic diagram of this research.

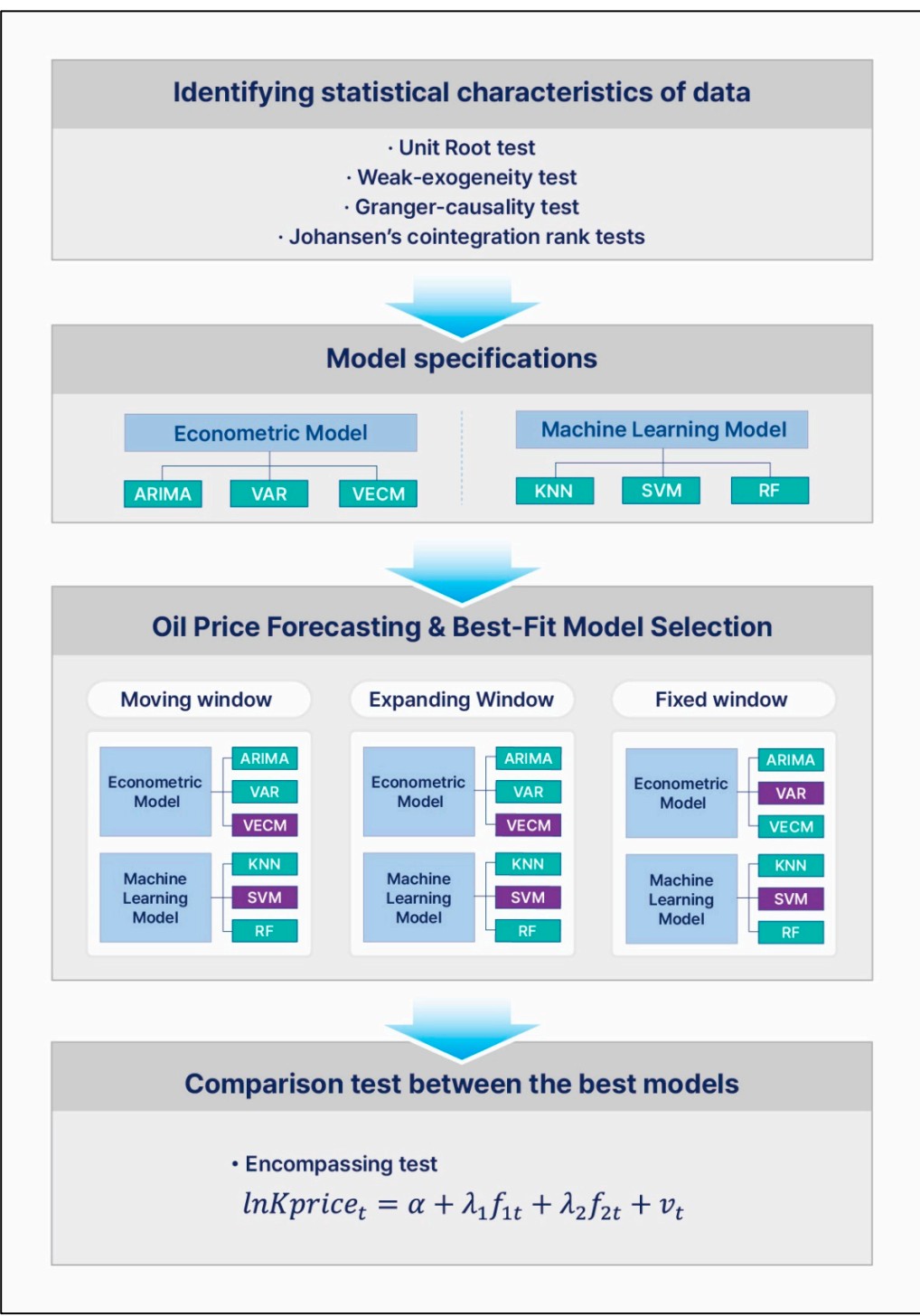

**Figure 1.** Schematic diagram of research.

## 2. Materials and Methods

### 2.1. Data

Table 1 presents a description of time series data for crude oil price indicators, divided into three categories: crude oil price, supply and demand factors, and supply chain factors. The dependent variable is crude oil import prices in South Korea, and the independent variables include Dubai crude oil spot prices, global balance (supply–demand), global strategic stocks, crude oil demand in South Korea, crude oil stocks in South Korea, and the Global Supply Chain Pressure Index (GSCPI). The crude oil consumption data in South

Korea, provided by Petronet [29], pertain to products processed from crude oil. In addition, as South Korea does not disclose strategic reserve oil data, we used commercial stocks disclosed by Petronet as an explanatory variable. We used monthly data covering the period from February 2008 to February 2022.

**Table 1.** Description of data for crude oil price indicators.

| Factors | Variables | Description | Source |
|---|---|---|---|
| Crude Oil Price | ln *Kprice* | Crude oil import prices in South Korea | Petronet |
| | ln *Dubai* | Global crude oil prices | Petronet |
| Supply–Demand | ln *Gbalance* | Global crude oil supply–global crude oil demand | KEEI data |
| | ln *Strategic* | Global strategic reserves | KEEI data |
| | ln *Kdemand* | Crude oil demand in South Korea | Petronet |
| | ln *Kstock* | Crude oil stocks in South Korea | Petronet |
| Supply Chain | ln *GSCPI* | Global Supply Chain Pressure Index | Newyorkfed.org |
| | | Crude oil import prices in South Korea | Petronet |

The data from Petronet and Newyorkfed are officially published, whereas the data from KEEI are sourced internally.

In this research, we would like to consider the Global Supply Chain Pressure Index (GSCPI) as a supply chain-related variable for the models. The Federal Reserve Bank of New York has reported the GSCPI since September 1997. According to the Federal Reserve Bank of New York, the GSCPI is designed to develop a parsimonious measure of global supply chain pressures that could be used to gauge the significance of supply constraints concerning economic outcomes. Additionally, as mentioned by Benigno et al. [30], the GSCPI captures factors that exert pressure on the global supply chain, both domestic and due to international trade. The GSCPI integrates global transportation costs such as the Baltic Dry Index (BDI), the Harpex index, and airfreight cost indices from the U.S. Bureau of Labor Statistics (BLS), as well as manufacturing indicators from Purchasing Managers' Index (PMI) surveys.

A number of studies have explored the relationship between the GSCPI and economic indicators. Benigno et al. [30] showed that recent inflationary pressures are closely linked to the GSCPI, especially at the level of producer price inflation in the United States and the euro area. Andriantomanga [31] also studied the impact of supply chain disruptions on inflation and monetary policy in sub-Saharan Africa. Their findings showed that increased supply chain pressures significantly impacted headline, food, and tradable inflation across a panel of 29 sub-Saharan African countries from 2000 to 2022. Laumer [32] investigated the impact of supply chain disruptions on consumer price inflation and found that a global supply chain shock significantly raised consumer price inflation in the US, the euro area, and the UK. Lastly, Kabaca and Tuzcuoglu [33] examined the effect of six supply shocks—labor supply, labor productivity, global supply chain, oil price, price mark-up, and wage mark-up shocks—on US headline inflation since the COVID-19 pandemic. The study revealed that the global supply chain and oil price shocks were the primary supply contributors to US inflation during the pandemic.

### 2.2. Model Specifications

This research employs three econometric models and three machine learning regression models: an autoregressive integrated moving average model (ARIMA), a vector autoregression model (VAR), a vector error correction (VECM) model, a k-nearest neighbor (KNN), a support vector machine (SVM), and a random forest (RF). We construct the econometric models based on the statistical characteristics of the data. To investigate the significant influence of the GSCPI on the forecast performance of crude oil import prices in South Korea, we employ the ARIMA model as the benchmark. Considering our focus on multivariate variables, we explore the VAR and VECM models. To construct the VAR model, we utilize the weak exogeneity test and the Granger causality test. For the VECM model, Johansen's cointegration rank test is employed. The machine learning models are

selected by the benchmark test to the deep neural network (DNN). The DNN model Long Short-Term Memory (LSTM) is established to provide decent forecast performance for the time domain forecasting problem. Meanwhile, the machine learning-based forecast models RF, KNN, and SVM that are selected in our research provide superior performance under the limited data quantity to learn. The benchmark test in Section 3 proves that the machine learning method shows outstanding forecast performance in our problem. Therefore, three machine learning methods, RF, KNN, and SVM, were selected. In the following section, the formulation and the background of each time series and machine learning model will be explained.

### 2.2.1. ARIMA Model

The first specifications of the autoregressive integrated moving average (ARIMA ($p$,$d$,$q$)) model in this research are as follows:

$$\Delta lnKprice_t = \theta_0 + \sum_{i=1}^{p} \varnothing_i \Delta lnKprice_{t-i} + \varepsilon_t - \sum_{k=1}^{q} \rho_j \varepsilon_{t-j}, \tag{1}$$

where $\Delta lnKprice_t$ represents the first differenced crude oil import price in South Korea, $\Delta lnKprice_{t-i}$ is the first differenced i th lags of $\Delta lnKprice_t$, and $\varepsilon_t$ denotes the stochastic error term, which is independently and identically distributed with a mean of zero and constant variance of $\sigma^2$.

### 2.2.2. VAR and VECM Model

In this research, we would like to consider multivariate variables to predict crude oil prices and investigate if the GSCPI has a statistically significant effect on crude oil prices. Thus, both a vector autoregression (VAR) model and a vector error correction (VECM) model are considered. The VAR model can be constructed both in level and first differences. In this research, we follow Sims et al.'s [34] work and estimate a VAR model in level. They argued that even if the variables are not stationary over time, using the variables in levels might be more appropriate than differencing. The specified VAR model in level form for this research is as follows:

$$
\begin{bmatrix}
lnKprice_t \\
lnDubai_t \\
lnGbalance_t \\
lnStrategic_t \\
lnKdemand_t \\
lnKstock_t \\
lnGSCPI_t
\end{bmatrix}
$$

$$
=
\begin{bmatrix}
\alpha_1 \\
\alpha_2 \\
\alpha_3 \\
\alpha_4 \\
\alpha_5 \\
\alpha_6 \\
\alpha_7
\end{bmatrix}
+
\begin{bmatrix}
\alpha_{11}^1 \alpha_{12}^1 \alpha_{13}^1 \alpha_{14}^1 \alpha_{15}^1 \alpha_{16}^1 \alpha_{17}^1 \\
\alpha_{21}^1 \alpha_{22}^1 \alpha_{23}^1 \alpha_{24}^1 \alpha_{25}^1 \alpha_{26}^1 \alpha_{27}^1 \\
\alpha_{31}^1 \alpha_{32}^1 \alpha_{33}^1 \alpha_{34}^1 \alpha_{35}^1 \alpha_{36}^1 \alpha_{37}^1 \\
\alpha_{41}^1 \alpha_{42}^1 \alpha_{43}^1 \alpha_{44}^1 \alpha_{45}^1 \alpha_{46}^1 \alpha_{47}^1 \\
\alpha_{51}^1 \alpha_{52}^1 \alpha_{53}^1 \alpha_{54}^1 \alpha_{55}^1 \alpha_{56}^1 \alpha_{57}^1 \\
\alpha_{61}^1 \alpha_{62}^1 \alpha_{63}^1 \alpha_{64}^1 \alpha_{65}^1 \alpha_{66}^1 \alpha_{67}^1 \\
\alpha_{71}^1 \alpha_{72}^1 \alpha_{73}^1 \alpha_{74}^1 \alpha_{75}^1 \alpha_{76}^1 \alpha_{77}^1
\end{bmatrix}
\begin{bmatrix}
lnKprice_{t-1} \\
lnDubai_{t-1} \\
lnGbalance_{t-1} \\
lnStrategic_{t-1} \\
lnKdemand_{t-1} \\
lnKstock_{t-1} \\
lnGSCPI_{t-1}
\end{bmatrix}
+ \dots
$$

$$
+
\begin{bmatrix}
\alpha_{11}^p \alpha_{12}^p \alpha_{13}^p \alpha_{14}^p \alpha_{15}^p \alpha_{16}^p \alpha_{17}^p \\
\alpha_{21}^p \alpha_{22}^p \alpha_{23}^p \alpha_{24}^p \alpha_{25}^p \alpha_{26}^p \alpha_{27}^p \\
\alpha_{31}^p \alpha_{32}^p \alpha_{33}^p \alpha_{34}^p \alpha_{35}^p \alpha_{36}^p \alpha_{37}^p \\
\alpha_{41}^p \alpha_{42}^p \alpha_{43}^p \alpha_{44}^p \alpha_{45}^p \alpha_{46}^p \alpha_{47}^p \\
\alpha_{51}^p \alpha_{52}^p \alpha_{53}^p \alpha_{54}^p \alpha_{55}^p \alpha_{56}^p \alpha_{57}^p \\
\alpha_{61}^p \alpha_{62}^p \alpha_{63}^p \alpha_{64}^p \alpha_{65}^p \alpha_{66}^p \alpha_{67}^p \\
\alpha_{71}^p \alpha_{72}^p \alpha_{73}^p \alpha_{74}^p \alpha_{75}^p \alpha_{76}^p \alpha_{77}^p
\end{bmatrix}
\begin{bmatrix}
lnKprice_{t-p} \\
lnDubai_{t-p} \\
lnGbalance_{t-p} \\
lnStrategic_{t-p} \\
lnKdemand_{t-p} \\
lnKstock_{t-p} \\
lnGSCPI_{t-p}
\end{bmatrix}
+
\begin{bmatrix}
\varepsilon_{Kpricet} \\
\varepsilon_{Dubait} \\
\varepsilon_{Gbalancet} \\
\varepsilon_{Strategict} \\
\varepsilon_{Kdemandt} \\
\varepsilon_{Kstockt} \\
\varepsilon_{GSCPIt}
\end{bmatrix}
\tag{2}
$$

where $a_{ij}^k$ i = 1, 2, 3, 4, 5, 6, 7, j = 1, 2, 3, 4, 5, 6, 7, and k = 1, 2, ..., $p$ denote the autoregressive coefficients, and $\varepsilon_{Kprice t}, \varepsilon_{Dubai t}, \varepsilon_{Gbalance t}, \varepsilon_{Strategic t}, \varepsilon_{Kdemand t}, \varepsilon_{Kstock t}$, and $\varepsilon_{GSCPI t}$ represent white noise disturbances with standard deviations of $\sigma_{Kprice}, \sigma_{Dubai}, \sigma_{Gbalance}, \sigma_{Strategic}, \sigma_{Kdemand}, \sigma_{Kstock}$, and $\sigma_{GSCPI}$, respectively.

In this research, the VECM model form is as follows:

$$
\begin{bmatrix} \Delta lnKprice_t \\ \Delta lnDubai_t \\ \Delta lnGbalance_t \\ \Delta lnStrategic_t \\ \Delta lnKdemand_t \\ \Delta lnKstock_t \\ \Delta lnGSCPI_t \end{bmatrix} = \begin{bmatrix} \delta_1 \\ \delta_2 \\ \delta_3 \\ \delta_4 \\ \delta_5 \\ \delta_6 \\ \delta_7 \end{bmatrix} + \begin{bmatrix} \gamma_{11} & \gamma_{12} & \gamma_{13} & \gamma_{14} & \gamma_{15} & \gamma_{16} & \gamma_{17} \\ \gamma_{21} & \gamma_{22} & \gamma_{23} & \gamma_{24} & \gamma_{25} & \gamma_{26} & \gamma_{27} \\ \gamma_{31} & \gamma_{32} & \gamma_{33} & \gamma_{34} & \gamma_{35} & \gamma_{36} & \gamma_{37} \\ \gamma_{41} & \gamma_{42} & \gamma_{43} & \gamma_{44} & \gamma_{45} & \gamma_{46} & \gamma_{47} \\ \gamma_{51} & \gamma_{52} & \gamma_{53} & \gamma_{54} & \gamma_{55} & \gamma_{56} & \gamma_{57} \\ \gamma_{61} & \gamma_{62} & \gamma_{63} & \gamma_{64} & \gamma_{65} & \gamma_{66} & \gamma_{67} \\ \gamma_{71} & \gamma_{72} & \gamma_{73} & \gamma_{74} & \gamma_{75} & \gamma_{76} & \gamma_{77} \end{bmatrix} \begin{bmatrix} lnKprice_{t-1} \\ lnDubai_{t-1} \\ lnGbalance_{t-1} \\ lnStrategic_{t-1} \\ lnKdemand_{t-1} \\ lnKstock_{t-1} \\ lnGSCPI_{t-1} \end{bmatrix} +
$$

$$
\begin{bmatrix} \varphi_{11}^1 & \varphi_{12}^1 & \varphi_{13}^1 & \varphi_{14}^1 & \varphi_{15}^1 & \varphi_{16}^1 & \varphi_{17}^1 \\ \varphi_{21}^1 & \varphi_{22}^1 & \varphi_{23}^1 & \varphi_{24}^1 & \varphi_{25}^1 & \varphi_{26}^1 & \varphi_{27}^1 \\ \varphi_{31}^1 & \varphi_{32}^1 & \varphi_{33}^1 & \varphi_{34}^1 & \varphi_{35}^1 & \varphi_{36}^1 & \varphi_{37}^1 \\ \varphi_{41}^1 & \varphi_{42}^1 & \varphi_{43}^1 & \varphi_{44}^1 & \varphi_{45}^1 & \varphi_{46}^1 & \varphi_{47}^1 \\ \varphi_{51}^1 & \varphi_{52}^1 & \varphi_{53}^1 & \varphi_{54}^1 & \varphi_{55}^1 & \varphi_{56}^1 & \varphi_{57}^1 \\ \varphi_{61}^1 & \varphi_{62}^1 & \varphi_{63}^1 & \varphi_{64}^1 & \varphi_{65}^1 & \varphi_{66}^1 & \varphi_{67}^1 \\ \varphi_{71}^1 & \varphi_{72}^1 & \varphi_{73}^1 & \varphi_{74}^1 & \varphi_{75}^1 & \varphi_{76}^1 & \varphi_{77}^1 \end{bmatrix} \begin{bmatrix} \Delta lnKprice_{t-1} \\ \Delta lnDubai_{t-1} \\ \Delta lnGbalance_{t-1} \\ \Delta lnStrategic_{t-1} \\ \Delta lnKdemand_{t-1} \\ \Delta lnKstock_{t-1} \\ \Delta lnGSCPI_{t-1} \end{bmatrix} + \tag{3}
$$

$$
\begin{bmatrix} \varphi_{11}^{p-1} & \varphi_{12}^{p-1} & \varphi_{13}^{p-1} & \varphi_{14}^{p-1} & \varphi_{15}^{p-1} & \varphi_{16}^{p-1} & \varphi_{17}^{p-1} \\ \varphi_{21}^{p-1} & \varphi_{22}^{p-1} & \varphi_{23}^{p-1} & \varphi_{24}^{p-1} & \varphi_{25}^{p-1} & \varphi_{26}^{p-1} & \varphi_{27}^{p-1} \\ \varphi_{31}^{p-1} & \varphi_{32}^{p-1} & \varphi_{33}^{p-1} & \varphi_{34}^{p-1} & \varphi_{35}^{p-1} & \varphi_{36}^{p-1} & \varphi_{37}^{p-1} \\ \varphi_{41}^{p-1} & \varphi_{42}^{p-1} & \varphi_{43}^{p-1} & \varphi_{44}^{p-1} & \varphi_{45}^{p-1} & \varphi_{46}^{p-1} & \varphi_{47}^{p-1} \\ \varphi_{51}^{p-1} & \varphi_{52}^{p-1} & \varphi_{53}^{p-1} & \varphi_{54}^{p-1} & \varphi_{55}^{p-1} & \varphi_{56}^{p-1} & \varphi_{57}^{p-1} \\ \varphi_{61}^{p-1} & \varphi_{62}^{p-1} & \varphi_{63}^{p-1} & \varphi_{64}^{p-1} & \varphi_{65}^{p-1} & \varphi_{66}^{p-1} & \varphi_{67}^{p-1} \\ \varphi_{71}^{p-1} & \varphi_{72}^{p-1} & \varphi_{73}^{p-1} & \varphi_{74}^{p-1} & \varphi_{75}^{p-1} & \varphi_{76}^{p-1} & \varphi_{77}^{p-1} \end{bmatrix} \begin{bmatrix} \Delta lnKprice_{t-p-1} \\ \Delta lnDubai_{t-p-1} \\ \Delta lnGbalance_{t-p-1} \\ \Delta lnStrategic_{t-p-1} \\ \Delta lnKdemand_{t-p-1} \\ \Delta lnKstock_{t-p-1} \\ \Delta lnGSCPI_{t-p-1} \end{bmatrix} + \begin{bmatrix} \varepsilon_{Kprice t} \\ \varepsilon_{Dubai t} \\ \varepsilon_{Gbalance t} \\ \varepsilon_{Strategic t} \\ \varepsilon_{Kdemand t} \\ \varepsilon_{Kstock t} \\ \varepsilon_{GSCPI t} \end{bmatrix}
$$

In this equation, at least one $\gamma_{ij}$ should not be zero because the VECM model can be expressed with a multivariate VAR model in first differences, augmented by the error correction term when $\gamma_{ij} = 0$.

### 2.2.3. K-Nearest Neighbor (KNN)

Abdella et al. [35] used a regression machine learning model when predicting airline ticket prices, and the results showed that the random forest model had excellent performance. KNN regression and SVR are commonly adopted machine learning models because those methods have decent clustering and forecast performance. KNN regression has a structure that is easy to interpret and is a powerful model in both classification and regression. SVR has little influence on outliers and enables regression analysis of nonlinear data by introducing a kernel function. These two models and the random forest model show superior performance under the limited amount of data available.

The KNN model is a supervised and localized learning algorithm used to build a regression model. The principle of KNN is to predict the value of a target variable, which is oil price in this manuscript, by finding the k-nearest neighbors of a given data point in the training dataset and using their average or median value as the predicted value. In the regression model, the value of K represents the value of adjacent neighbors. A small K value can reflect the local characteristics of the data excessively, resulting in overfitting of the model. Conversely, a larger K value tends to regularize the forecast model.

The KNN model first calculates the distance between the data points in the training dataset and the data point for which we want to forecast the target variable. The most commonly used distance metric is the Euclidean distance. Once the distances are calculated, the algorithm identifies the k-nearest neighbors, where k represents the number of neighbors.

Finally, the algorithm computes the predicted value for the target variable by taking the average or median of the target values of the k-nearest neighbors.

The KNN model used in this manuscript is the one suggested by Kantz [36], as presented in Equations (4) and (5):

$$lnKprice_t = \sum_{i=1}^{t-1} W(x, x_i)x_i \tag{4}$$

where $lnKprice_i$ is the label oil category's Consumer Price Index, $W$ is the weight factor, and $x_i = [Kprice_i, Dubai_i, Gbalance_i, Strategic_i, Kdemand_i, Kstock_i, GCPI_i]$

$$W(x, x_i) = \frac{exp(-D(x, x_i))}{\sum_{i=1}^{k} exp(-D(x, x_i))} \tag{5}$$

where $D$ is the distance between the query point x and the *i*-th case xi of the training feature.

The performance of the KNN model is dependent on the value of the hyper-parameters; hence, this manuscript adopts the grid search method that aims to find the optimal parameters for the KNN model proposed by Ambesange [37]. The hyper-parameters of the KNN regressor defined by the result of the grid search are presented in Table 2.

**Table 2.** Selected hyper-parameters for KNN regressor.

| Hyper-Parameters | Value |
|:---:|:---:|
| K (Number of Neighbors) | 5 |
| W (Weight) | Euclidean Distance |

### 2.2.4. Support Vector Machine (SVM)

Support vector machine (SVM) regression is a supervised learning algorithm used to generate a regression model. In SVM regression, the goal is to find the optimal weight 'w_SVM' that maximizes the margin from the closest data points at the decision hyperplane. The margin denotes the distance between the hyperplane and the closest data points, and the hyperplane with the largest margin is considered the best fit. In SVM regression, the choice of kernel function plays a crucial role in determining the quality of the fit. Linear, polynomial, and radial basis function (RBF) kernels are the most commonly used kernel functions in SVM regression. The training process in SVM regression involves finding the optimal values for the model parameters, including the kernel function, regularization parameter, and kernel function parameters. This manuscript adopts the Lagrangian multiplier to find the optimal values.

The SVM utilized in this manuscript is based on the proposal by Fan [38]. The SVM forecasts linear regression by using the weighing factor wi and the slack variable, as presented in Equation (6):

$$lnKprice_i = w_{svm}\ln(x_i) + \beta \tag{6}$$

where $Kprice_i$ is the label, $x_i$ is the i-th training features, $w_{svm}$ is the weight, $\beta$ is the intercept of the linear function.

The grid search method that optimizes the hyper-parameter was conducted, as suggested by Paul [39]. The margin is an area that does not include any data. The regression model is to learn to encompass as many data as possible within the margin. The hyper-parameters of the SVM regressor are defined as shown in Table 3.

**Table 3.** Selected hyper-parameters for SVM.

| Hyper-Parameters | Value |
|:---:|:---:|
| Regularization Parameter | 5 |
| Width of Margin Epsilon ($\varepsilon$) | 0.01 |
| Weight ($w_{SVM}$) | Distance |

### 2.2.5. Random Forest (RF)

The random forest regressor is a meta-predictor that first builds several classifying decision trees on sub-samples of the training dataset. It is an ensemble learning method that combines multiple decision trees to increase prediction accuracy. The random forest regressor generates many decision trees, each of which is trained on a random subset of the training data and a random subset of the features. During training, the decision trees are built in such a way that they split the data based on the feature that provides the most information gain, as presented in Figure 2.

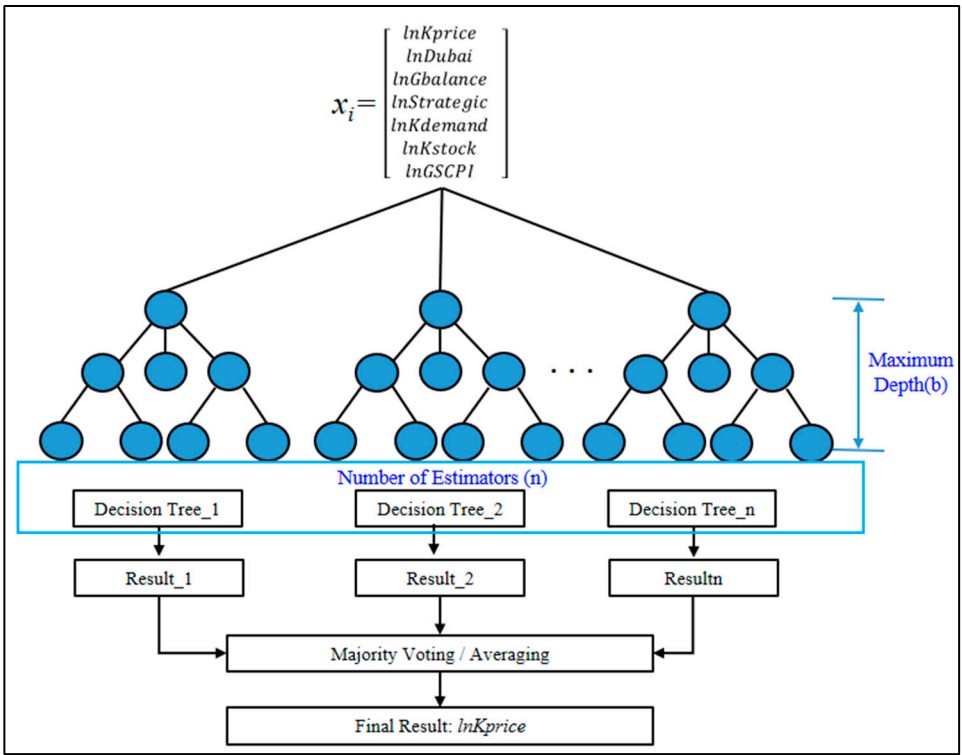

**Figure 2.** Random forest structure.

Each tree in the forest predicts an output value during prediction, and the final output is the average of all the individual tree forecasts. This forecasting process helps to reduce overfitting, as the combination of multiple trees can smooth out the noise in the data and provide more robust predictions.

Random forest [40] is a supervised learning algorithm that uses the decision tree structure, as presented in Equations (7) and (8).

$$x_i = [Kprice_i, Dubai_i, Gbalance_i, Strategic_i, Kdemand_i, Kstock_i, GCPI_i,] \tag{7}$$

where $x_i$ is i-th of training subset vector.

$$\hat{P}(y = 1|x_i) = \frac{1}{b}\sum_{i=1}^{b} \hat{P}_{l_k(x_i)(y=1)} \tag{8}$$

where b represents the depth of tree, P is Bayesian probability, $x_i$ denotes the training subset, y is the forecast target lnKprice.

The hyper-parameters for random forest regression are defined in Table 4. The grid search method is applied to find the optimal hyper-parameter that yields the best forecast precision for the random forest model. As a result, the proposed random forest model is composed of 50 trees with 100 depths.

**Table 4.** Selected hyper-parameters for random forest (RF).

| Hyper-Parameters | Value |
|---|---|
| Number of Estimator ($n$) | 50 |
| Maximum Depth (b) | 100 |
| Criterion | Squared Error |

## 3. Results

To investigate whether the supply–demand and supply chain factors are weak exogenous variables, we conducted a weak exogeneity test utilizing the standard Wald test. The null hypothesis of a weak exogenous variable was rejected at 1% level for $\ln Kprice$, $\ln Dubai$, $\ln Gbalance$, $\ln Strategic$, $\ln Kdemand$, and $\ln GSCPI$, and for $\ln Kstock$ at the 5% level. These results imply that supply–demand and supply chain factors react to disequilibrium in the long run and could improve the accuracy of predicting the $\ln Kprice$. Table 5 below shows the results of the weak exogeneity test. In the case of a larger VAR (n > 2), the Granger causality restriction implies a weak exogeneity form. The results of the Granger causality test based on the VAR and VECM models align with those of the weak exogeneity test. In both models, tests 1, 2, 3, 5, 6, and 7 reject the null hypothesis at the 1% significance level, while test 4 does so at the 10% and 5% levels. For the test, group 1 includes variables from $\ln Kprice$ to $\ln GSCPI$, each corresponding to a specific test (e.g., $\ln Kprice$ for test 1, $\ln Dubai$ for test 2, $\ln GSCPI$ for test 7). Group 2 comprises the remaining six variables.

**Table 5.** The results of weak exogeneity test.

| Variable | $\chi^2$ | $Pr>\chi^2$ |
|---|---|---|
| $\ln Kprice$ | 195.18 | <0.0001 *** |
| $\ln Dubai$ | 30.51 | <0.0001 *** |
| $\ln Gbalance$ | 40.99 | <0.0001 *** |
| $\ln Strategic$ | 13.95 | 0.0075 *** |
| $\ln Kdemand$ | 29.11 | <0.0001 *** |
| $\ln Kstock$ | 12.08 | 0.0168 ** |
| $\ln GSCPI$ | 19.51 | 0.0006 *** |

The last column entry is the *p*-value of the null hypothesis of a weak exogenous variable. The asterisks **, and *** indicate the null hypothesis can be rejected at the 0.05, and 0.01 levels, respectively.

Since the variables are non-stationary over time and all have a single unit root, a Johansen's cointegration rank test is conducted to ascertain the presence of a long-run equilibrium relationship between the variables. Table 6 indicates the results of Johansen's cointegration test. Both trace and maximum eigenvalue tests fail to reject the null hypothesis of four cointegration vectors at the 5% level. The below table indicates the long-run equilibrium relationship in the VECM model, which consists of the long-run parameter β and the adjustment coefficient α with $\ln Kprice$ normalized.

Based on the results of the weak exogeneity test and Johansen's cointegration rank tests, we constructed the VAR model and the VECM model. Further, we employed the KNN, SVM, and RF models to forecast the target variable, lnKprice. In principle, the proposed three machine learning models are supervised models and require sufficient subsets of training. The number of data in the training set is one hundred and seventy in this research, and it is not sufficient to build a machine learning forecast model. Therefore, the fine-time interval data might enhance the performance of models. In addition, the optimization method's hyperplane parameters also make a difference in the forecast performance. This research adopts the grid search method that experiments with the forecast effects due to the combination of the hyperplane parameters of the model. In particular, it is necessary to define the weight (wSVM) that maximizes the margin of the forecast model in the SVM model. The optimization method is involved in the weight decision process. This research adopts the Lagrangian multiplier to define the weight that aims to have the maximum margin, but other optimization methods, such as gradient descent, the genetic algorithm,

the particle swarm method, and steal annealing, can be applied to optimize the SVM model based on the characteristics of the dataset.

**Table 6.** Johansen's cointegration rank tests.

| **Trace Test** | | | |
|---|---|---|---|
| $H_0 : Rank = r$ | $H_0 : Rank > r$ | Trace Statistics | Pr > Trace |
| 0 | 0 | 568.2114 | <0.0001 |
| 1 | 1 | 171.0168 | <0.0001 |
| 2 | 2 | 100.6094 | <0.0001 |
| 3 | 3 | 52.807 | 0.0014 |
| 4 | 4 | 21.6404 | 0.1039 |
| **Maximum Eigenvalue Test** | | | |
| $H_0 : Rank = r$ | $H_0 : Rank = r + 1$ | Max Statistics | Pr > Maximum |
| 0 | 1 | 397.1946 | <0.0001 |
| 1 | 2 | 70.4074 | <0.0001 |
| 2 | 3 | 47.8024 | 0.0004 |
| 3 | 4 | 31.1666 | 0.0046 |
| 4 | 5 | 12.6967 | 0.2474 |

We considered three window schemes, moving window, expanding window, and fixed window, to compare the forecasting performance of the resulting models. In the moving window, the model predicts based on 119 monthly data. We measure the first one-step-ahead forecasts using the first 119 observations (from February 2008 to December 2017). For the second forecast values, we drop the very first observation (February 2008) and include the 120th datum (January 2018). In this case, the size of the window is fixed at 119. For the expanding window, we used all the data available to estimate the one-step-ahead forecast values. The dataset used to estimate the 120th forecast value (January 2018) is the same as the moving window scheme. However, to estimate the 121st prediction (February 2018), the expanding window scheme incorporates all the data from the 1st to the 120th (from February 2008 to January 2018), whereas the moving window dropped the very first datum (February 2008). In other words, the size of the expanding window increases by one as time goes by. Lastly, for the fixed window scheme, we used 119 in-sample data points (from February 2008 to December 2017) to estimate the 50 forecast values (from January 2018 to February 2022).

To assess the predictive performance of the forecast models, we utilized the root mean square error (RMSE) and the mean absolute percentage error (MAPE) of each forecasting model. According to Table 7, for the time series model, the VAR and the VECM models outperform the ARIMA model under moving window and expanding window schemes. This result implies that the crude oil supply–demand factors and supply chain factors are useful in improving the forecasting performance of the crude oil import price in South Korea because the VAR and VECM models are suitable methods to forecast the target multivariable that mutually influences each. Additionally, the VECM model has the smallest RMSE under the moving window scheme and has the smallest RMSE and MAPE under the expanding window scheme. This result implies the VECM model can be an appropriate method when the large time series data are sufficient to find the equilibrium among the occupied multivariable. For the fixed window scheme, the table denotes that the VAR model outperforms the ARIMA and VECM models. These results suggest that the VECM model is a superior model when the amount of data is relatively limited. Therefore, the VECM and the VAR models are recommended in most situations to forecast the effect on crude oil by the Global Supply Chain Pressure Index).

**Table 7.** Comparison of machine learning and deep learning.

| Performance of AI | RMSE | MAPE |
|---|---|---|
| DNN | 7.3421 | 1.788 |
| KNN | 0.2335 | 4.2130 |
| SVR | 0.11048 | 1.6716 |
| Random Forest | 0.1221 | 1.8657 |

The performance comparison Is conducted between the deep neural network (DNN) and the machine learning models, as presented in Table 7. The superior forecast performance of the machine learning method is valid under the limited data quantity. It was empirically established that machine learning methods would perform better than deep learning for numerical data. As a result of verification, the results of comparing the DNN and machine learning methods without any special tuning were as follows. Among the total data, those from the period from 2008 to 2017 were used as learning data, and the data from 2018 to 2022 were used as testing data.

For the machine learning models, Table 8 shows that the random forest model outperforms under the expanding window and fixed window schemes. The SVM model has the best performance at the fixed window scheme and is slightly better than KNN under the moving and expanding window. The SVM and the RF models have a robust forecast performance when the training datasets are insufficient and unbalanced compared to the KNN model. The random forest model basically consists of the ensemble model, so it shows a decent performance when the data are not sufficient for machine learning. Our problem is more explainable with a hyper-parameter optimized by the Lagrangian multiplier. This finding aligns with a previous study by Keerthan [41], which also shows the superiority of the SVM forecast model for oil price prediction that has a similar structure to our dataset.

**Table 8.** One-step-ahead crude oil import price in South Korea forecasting comparison by moving and expanding window schemes, and fifty-step-ahead crude oil import price in South Korea forecasting comparison under fixed window scheme.

| Moving Window Scheme | | | |
|---|---|---|---|
| Models | | RMSE | MAPE |
| Time Series Model | ARIMA(1, 1, 1) | 0.07192 | 1.195155 |
| | VAR(1) | 0.04608 | 0.790037 |
| | VECM(1) | 0.0451 | 0.794936 |
| Machine Learning Model | KNN | 0.33586 | 6.449434 |
| | SVM | 0.25460 | 6.449061 |
| | RF | 0.34308 | 6.631965 |
| Expanding Window Scheme | | | |
| Models | | RMSE | MAPE |
| Time Series Model | ARIMA(1, 1, 1) | 0.07088 | 1.197634 |
| | VAR(1) | 0.04921 | 0.809486 |
| | VECM(1) | 0.04791 | 0.798104 |
| Machine Learning Model | KNN | 0.26189 | 4.79626 |
| | SVM | 0.18020 | 4.79052 |
| | RF | 0.11115 | 1.745939 |
| Fixed Window Scheme | | | |
| Models | | RMSE | MAPE |
| Time Series Model | ARIMA(1, 1, 1) | 0.26162 | 4.671115 |
| | VAR(1) | 0.25478 | 4.656029 |
| | VECM(1) | 0.26432 | 4.817335 |
| Machine Learning Model | KNN | 0.2335 | 4.213045 |
| | SVM | 0.11048 | 1.671631 |
| | RF | 0.11339 | 1.865766 |

Interestingly, under the moving window and expanding window, the time series model seems to have better prediction performance than the machine learning model.

However, under the fixed window scheme, the machine learning models outperform the conventional time series models. This could imply that the prediction performance of machine learning and time series models might vary depending on the prediction range.

Regarding the sensitivity analysis, the result of the sensitivity analysis is shown in Table 9. The windows of the fixed, moving, and expanding schemes are varied to analyze the forecast performance of the machine learning method. Three window sizes, 99, 119, and 139, are selected and experimented with. Based on the sensitivity test results of the experiment with the learning data size, the forecast tendency of the machine learning method is maintained. Therefore, it can be concluded that this forecast scheme has robustness for this crude oil price prediction using the relation of the GSCPI.

**Table 9.** Sensitivity analysis.

| Window Type | ML(Error) | Window Size | | |
|---|---|---|---|---|
| | | 99 | 119 | 139 |
| Expanding Widow | SVM(RMSE) | 0.2788 | 0.1802 | 0.1713 |
| | KNN(RMSE) | 0.3804 | 0.2618 | 0.1903 |
| | RF(RMSE) | 0.1168 | 0.1111 | 0.0599 |
| Moving Window | SVM(RMSE) | 0.3806 | 0.2546 | 0.1900 |
| | KNN(RMSE) | 0.3807 | 0.3358 | 0.1900 |
| | RF(RMSE) | 0.3781 | 0.2568 | 0.1837 |

Based on the preliminary comparison results, we conducted the encompassing test to compare the forecast values of the conventional time series model with those of the machine learning model. The below is the equation for the test:

$$lnKprice_t = \alpha + \lambda_1 f_{1t} + \lambda_2 f_{2t} + v_t \tag{9}$$

where $lnKprice_t$ is the real value of the crude oil import price in South Korea, $f_{1t}$ is the forecast values from the time series model, $f_{2t}$ is the forecast values from the machine learning model, $\lambda_i$ are the coefficients of i th forecast, and $v_t$ is the error term.

Table 10 denotes that we could reject both the null ($H_0 : \lambda_1 = 0$) and alternative ($H_1 : \lambda_2 = 0$), which means the combined (or weighted) forecasts with $f_{1t}$ and $f_{2t}$ provide a better forecast at the 5% level. That is, for the moving window and expanding window schemes, the combined VECM forecasts and SVM forecasts would provide better forecast information. Further, under the fixed window scheme, the combined VAR and SVM prediction values could improve the forecasting accuracy.

**Table 10.** Encompassing test by moving, expanding, and fixed window scheme.

| Moving Window Scheme | | |
|---|---|---|
| Models | *t*-value | Pr > t |
| VECM(1) | 56.73 | <0.0001 |
| SVM | 3.3 | 0.0018 |
| **Expanding Window Scheme** | | |
| Models | *t*-value | Pr > t |
| VECM(1) | 43 | <0.0001 |
| SVM | 3.94 | 0.0003 |
| **Fixed Window Scheme** | | |
| Models | t-value | Pr > t |
| VAR(1) | 2.03 | 0.0475 |
| SVM | 15.14 | <0.0001 |

The last column entry is the *p*-value of the null hypothesis of a weak exogenous variable.

## 4. Discussion

As the importance of the role of supply chain risk management for strategic materials, especially those that are 100% import-dependent, has increased, we incorporated the supply chain-related variable into the crude oil import price forecasting model. This research investigates whether the supply chain factor, represented by the GSCPI, could significantly influence the improvement in the forecast performance of crude oil import prices in South Korea. We conducted the weak exogeneity test to see if the GSCPI is a weak exogenous variable. The null hypothesis ($H_0 : \ln GSCPI = weak\ exogenous\ variable$) was rejected at the 1% level, which implies that the GSCPI might improve the predicting accuracy of the $\ln Kprice$. Furthermore, we compared the forecasting performance of the VAR and VECM models, including the GSCPI with the ARIMA model, and found that the models with the GSCPI outperformed the models without the GSCPI in predicting the crude oil import price in South Korea. Based on these results, we propose that monitoring variables related to supply chain disruptions, such as the GSCPI, could be effective in stabilizing domestic prices and establishing long-term sustainable supply chain or energy policies. For instance, the South Korean government is currently seeking to enact a basic bill to support supply chain stabilization for economic security. This proposed bill includes the selection of economic security items and the operation of an early warning system to proactively identify and respond to supply chain risks. Building on the findings of this research, the authors suggest that early warning systems for crude oil should include monitoring of the GSCPI. Moreover, these implications are applicable to countries facing conditions similar to South Korea, particularly those heavily dependent on 100% oil imports.

In this research, a novel hybrid factor-based approach is proposed. We compared the forecasting performance of time series models ARIMA, VAR, and VECM, as well as the machine learning models KNN, SVM, and RF, using RMSE and MAPE under three different window schemes. As shown in Table 8, for time series models, the VECM model outperforms the ARIMA and VAR models under the moving and expanding window schemes, while the VAR model outperforms the ARIMA and VECM models under the fixed window scheme. For the machine learning model, the SVM model has the smallest RMSE and MAPE under all three window schemes. Based on these preliminary comparison results, we conducted the encompassing test to compare the forecast values of traditional time series models with those of machine learning models. Interestingly, the results of the encompassing test indicated that combining forecasts from time series models and machine learning models provided a better forecast. These findings are consistent with the previous research, which showed better prediction performance of proposed hybrid models than their counterparts (Safari and Davallou [23]; Zhang et al. [27]; Abdollahi [28]; He et al. [42]; Ning et al. [43]).

This research is meaningful in that it may serve as a foundation for the development of future oil price prediction models by examining whether supply chain-related variables are important factors in oil price prediction and what methodologies can be applied to enhance the forecasting accuracy of oil prices. Based on this, it is expected that more sophisticated forecasting models can be developed in future studies. In addition, it will be necessary to continue to discover supply chain-related variables such as the GSCPI for oil price prediction.

## 5. Conclusions

In this research, we aim to offer valuable insights for policymakers tasked with establishing a stable supply and demand strategy for strategic commodities and a national price stability strategy. Given South Korea's 100% dependence on crude oil imports, forecasting crude oil prices is crucial. This research emphasizes the importance of monitoring supply chain-related variables to enhance the predictive performance of crude oil prices, proposing a hybrid factor-based forecasting approach.

Nevertheless, there are a few considerations to solidify this approach. Firstly, this research evaluated the forecast performance of three representative machine learning-based

regression methods to determine if these could enhance the forecasting performance of the crude oil import price in South Korea. We analyzed the performance of the time windows of SVM, RF, and KNN. This research contributes by suggesting the appropriate hyper-parameter to build the machine learning model-based analysis framework. The grid search method was employed to find the optimal value that boosts the forecast performance of machine learning models. In addition, the weight of the SVM is suggested by the Lagrangian multiplier-based optimization. The machine learning-based estimator presents the forecast excellence of the fixed window-based forecast and the encompassing experiments.

Meanwhile, further studies can be conducted to optimize the SVM's parameter selection. Investigating different optimization methods, such as gradient descent, the genetic algorithm, and the particle swarm method, could help define the weighting factor more effectively. In addition, the machine learning method's forecast performance is also required to be studied when the time interval of the learning feature is finer than in this research. If the number of learning features is insufficient, then the augmentation or the replication of the time series data can be examined. Furthermore, other real-time-based estimators, such as the Long Short-Term Memory (LSTM) model, could be helpful for oil price prediction. LSTM is the one type of recurrent neural network model that has strong forecast performance in real-time variant data.

The data of the crude oil price is gathered from Petronet, but the oil price data can be gathered in identical forms from other sources such as Bloomberg energy and Wall Street Journal market data [44,45]. Global balance, Strategic Reserves, and GSCPI are global indexes, so their data can be obtained from various sources (e.g., Bloomberg and the Organization of the Petroleum Exporting Countries (OPEC) [44,46]) in similar forms. The limitation of this paper is that it only targeted the Korean market, which is described by Kprice, Kdemand, and Kstock, so future works can look into applying our model to other markets, for example, Europe, China, and the United States.

The change in the measurement frequency of the original data source that represents the relation between the crude oil price and GSCPI could be a valuable topic to be studied further in the future. If sufficient data, enough to apply the deep learning model, is reserved, then it can be used to predict the future time domain tendency of the crude oil price. Also, the recent generative AI foundation model has the potential to make a general AI model that can answer the projection of the crude oil price due to the GSCPI and other parameters' conjectures.

Lastly, as Livieris [47] mentioned, AI methods do not guarantee better forecasting performance in all cases. In this research, we found that the forecasting accuracy of traditional econometric models outperformed that of machine learning methods in the case of one-step-ahead oil price forecasting. While we considered three window schemes—moving, expanding, and fixed window—for the robustness of the estimates, it would be interesting to conduct a forecasting horizon sensitivity test, as it could impact the forecasting performance. Furthermore, it might be meaningful to analyze specific cases where the machine learning method demonstrates higher accuracy. Future research should focus on specific cases that contribute to increasing the prediction accuracy of econometric models, machine learning models, or hybrid models.

**Author Contributions:** Conceptualization, J.J. and E.L.; methodology, J.J.; software, J.L.; validation, J.J., J.L. and S.K.; formal analysis, J.J.; investigation, U.K.; resources, E.L.; data curation, J.J.; writing—original draft preparation, J.J.; writing—review and editing, S.K.; visualization, J.L.; supervision, E.L.; project administration, U.K.; funding acquisition, E.L. All authors have read and agreed to the published version of the manuscript.

**Funding:** This work was supported by the Korea Maritime Institute (KMI) grant funded by the Korean government under project title "A study on impact analysis and response measures of global supply chain risk".

**Data Availability Statement:** Data is contained within the article.

**Conflicts of Interest:** The authors declare no conflict of interest.

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
