# Peer review of "A Supply Chain-Oriented Model to Predict Crude Oil Import Prices in South Korea Based on the Hybrid Approach"

_sustainability, doi:10.3390/su152416725_

Round 1

Reviewer 1 Report

Comments and Suggestions for Authors

Dear Author,

I have completed my review of your manuscript titled "A Supply Chain-Oriented Model to Predict Crude Oil Import Prices in South Korea Based on the Hybrid Approach." The relevance of the topic in the current global economic scenario is well-understood, and your attempt to integrate supply chain variables into the price forecasting model is commendable. However, some areas of the manuscript could be significantly improved to meet the journal's standards for publication. Here are my detailed comments:

Clarification of Research Originality and Contribution: The manuscript does not sufficiently highlight the innovative aspects of your research. You are encouraged to further define your research motivation, articulate the specific questions you are addressing, and the anticipated contributions to the existing body of knowledge. A clearer exposition of how the GSCPI's inclusion advances the forecasting models for crude oil prices could bolster the perceived novelty of your work.

Methodological Framework Diagram: To aid reader comprehension, it would be beneficial to include a diagrammatic representation of your methodological framework. A visual flowchart or framework could illustrate the process from data collection through to model application and result interpretation. This will not only improve readability but also give a quick overview of your research process, highlighting the integration of time series and machine learning models.

Innovation in Research Methods: Please elucidate on the innovative aspects of your methodology. Explain how your approach differs from or improves upon existing models in the literature. Describe any novel algorithms, processes, or analytical techniques you employed and discuss their theoretical value. Articulating this will demonstrate the methodological robustness and theoretical advancement your paper proposes.

Implications for Other Countries: While your research focuses on South Korea, its findings could be of international interest. Expand your discussion to include potential implications or applications of your model to other nations' oil price forecasting, like China, the United States, and the European Union. This broadened perspective would be beneficial for policy makers and researchers interested in the application of your model to different geopolitical and economic contexts.

Your manuscript has the potential to make a significant contribution to the field of supply chain management and economic forecasting. I encourage you to address these comments in your revised manuscript. I look forward to seeing your research developed further, as it represents an important inquiry into the impacts of global risks on supply chain performance and commodity pricing.

Thank you for the opportunity to review your work.

Comments on the Quality of English Language

 Moderate editing of English language required

Reviewer 2 Report

Comments and Suggestions for Authors

Congratulations to the authors. In my opinion, the work can be published in a journal.

Pay special attention to the formatting of your article. Headings and chapter titles do not comply with MDPI requirements. Similarly, captions under figures and table captions. This must be done absolutely.

In order to strengthen the article, I propose:

In abstract, you must emphasize the originality and novelty of this article,

In the introduction, include a section that further describes the originality of the article

In the introduction, add a section that shows how your proposed model differs from the others already described

Number the patterns

As it stands, the research methodology is not clear - please correct this section

There are no research questions on which the article focuses

Your article has no conclusions, add this chapter. In conclusions, refer to specific tables and calculations

Reviewer 3 Report

Comments and Suggestions for Authors

This review paper investigated “A Supply Chain-Oriented Model to Predict Crude Oil Import Prices in South Korea Based on the Hybrid Approach.” This manuscript discusses a supply chain-oriented variable that can significantly impact the forecasting of crude oil import prices in South Korea. Our findings indicate that models with the GSCPI have gained attention. Although the title seems to be interesting, it cannot be published in "Sustainability" in its current form. The main concerns are summarized as follows:

1. Abstract - It needs to include the process of approaching and projecting the outcomes, such as performance and reliability.

2. Introduction - The authors must highlight core achievements and main contribution outlooks. Still, this section needs to extend advanced achievements. The research roadmap is missing in this section as well.

3. Literature -  The literature section is not convincing in highlighting research gaps. This paper lacks a corresponding literature review.

4. Methods – The superiority and complexity of the proposed machine learning are not discussed. Why did the authors not consider other AI-based algorithms to validate or compare with this method?

5. Results & Analysis - The experimental analysis is not profound enough. Also, Authors should consider more sensitivity analysis to clarify research outcomes.

6. Results & Analysis - The differences I see in many entries in the table are very small. The statistical tests are done at the global level and not instance per instance.

7. Discussion - Add this section before the conclusion. Moreover, the authors should mention the main gaps and limitations they have filled out and discuss managerial implications.

8. Conclusion - The data source of this paper is single, so the application value of the conclusions is low. Also, future directions should be discussed.

*** Minor Comments ***

-          Be consistent in mathematical model formatting.

-          please insert the references in alphabetical order in the list of references. Many of the references are outdated.

-          What is the research flowchart framework to indicate to us about the overview of this research?

additional comments:

1. What is the main question addressed by the research?

- The main question addressed by this research is "Can a supply chain-oriented model accurately predict crude oil import prices?"

- The findings of this research could have important implications for various stakeholders in the oil industry. For instance, it could help oil importing countries to better budget for their energy needs, and oil companies to optimize their supply chain operations.

2. Do you consider the topic original or relevant in the field? Does it

address a specific gap in the field?

- you should consider the novelty and relevance of the research topic, as well as its contribution to filling a gap in the existing literature:

- Consider the relevance of the research topic in the current context.

- Identify the specific gap in the field that the research addresses. For example, "The research addresses a specific gap in the field by proposing a model that incorporates supply chain factors, which have been largely overlooked in existing prediction models."

3. What does it add to the subject area compared with other published

material?

- By incorporating supply chain factors, this research provides a more comprehensive understanding of the factors influencing crude oil import prices. This could potentially improve the accuracy of future predictions and inform strategies for managing price volatility."

- Compare this research with other published material in the field.

4. What specific improvements should the authors consider regarding the methodology? What further controls should be considered?

- The authors could consider incorporating more granular data on supply chain factors, such as regional variations in production, transportation, and storage costs. This could potentially improve the accuracy of the model.

5. Are the conclusions consistent with the evidence and arguments presented

and do they address the main question posed?

- The authors conclude that a supply chain-oriented model can accurately predict crude oil import prices. This conclusion is supported by the evidence presented in the paper, which shows a strong correlation between supply chain factors and crude oil prices. However, the authors do not fully address the potential impact of geopolitical events on crude oil prices, which could limit the accuracy of their model.

6. Are the references appropriate?

- it suggests to considered several updated refrences.

7. Please include any additional comments on the tables and figures.

- Highlight how the findings in the tables and figures contribute to the broader theoretical understanding or practical implications in the field.

Reviewer 4 Report

Comments and Suggestions for Authors

Texts of subchapters 2.3. Supply chain factors and 2.4 Methods are the same (Rows 136-160 and Rows 163-187). Moreover, the title of the subchapter 2.4 Methods and chapter 3. Methods are the same. I suggest deleting subchapter 2.4 (Rows 163-187).

In Table 2 please correct Euclidian to Euclidean

The author writes little about the limits of the research. It describes that 'machine learning models do not always outperform traditional time series models', but does not mention, for example, that both exogeneity and causality play different roles in modeling and forecasting but exogeneity is neither necessary nor sufficient for causality. These considerations may modify the results of forecasts, therefore the uncertainty factor may remain high. Moreover, weak exogeneity in a time series regression with many controls may produce very large biases and can even lead to inconsistency of the least squares (OLS) estimator.

Comments on the Quality of English Language

There are some typos, that need to be corrected.

Round 2

Reviewer 1 Report

Comments and Suggestions for Authors

As a peer reviewer for the submitted manuscript "A Supply Chain-Oriented Model to Predict Crude Oil Import Prices in South Korea Based on the Hybrid Approach," I have thoroughly evaluated its content, methodology, and overall contribution to the field. Below is my review and recommendation:

Originality and Relevance:

The manuscript addresses a significant and timely issue – the influence of supply chain factors on crude oil prices, particularly in the context of South Korea. Given the current global economic landscape, marked by events like the COVID-19 pandemic, the Russia-Ukraine war, and the U.S.-China trade dispute, this research is highly relevant and contributes to a gap in existing literature​​.

Methodological Rigor:

The study employs a comprehensive approach, integrating both time series and machine learning models (ARIMA, VAR, VECM, KNN, SVM, RF) to enhance the predictive accuracy of crude oil prices. This hybrid methodology not only demonstrates the authors’ deep understanding of the subject but also showcases an innovative approach to forecasting​​.

However, I recommend a more detailed explanation of the selection criteria for these models and their specific suitability for the study to strengthen the methodological section.

Data and Analysis:

The authors have identified a wide range of determinants, including supply, demand, financial, commodity market, speculative, and political factors, which provide a comprehensive foundation for their forecasting models. This holistic approach is commendable as it accounts for the multifaceted nature of oil pricing​​.

I suggest expanding on how these factors were quantified and integrated into the models, as well as providing more details about the data sources, to enhance the transparency and reproducibility of the research.

Findings and Implications:

The manuscript's findings that models incorporating the Global Supply Chain Pressure Index (GSCPI) outperform those without it are significant. This highlights the importance of considering supply chain-related variables in crude oil price prediction, offering valuable insights for policymakers and industry practitioners​​.

The study would benefit from a deeper discussion on the policy implications of these findings, especially in the context of South Korea’s heavy reliance on crude oil imports.

Presentation and Structure:

The manuscript is well-structured and presents its arguments in a clear and logical manner. The abstract provides a succinct overview of the study, and the introduction effectively sets the stage for the research.

Recommendation:

I recommend acceptance of this manuscript, subject to minor revisions. The suggested improvements mainly pertain to expanding on methodological details and data sources, as well as elaborating on the policy implications of the findings.

This manuscript makes a valuable contribution to the field of supply chain management and crude oil price forecasting. Its relevance in the current global economic scenario and its methodological robustness are particularly noteworthy.

Comments on the Quality of English Language

Can be improved

Reviewer 3 Report

Comments and Suggestions for Authors

The authors have addressed my concerns, and I fully support this version of the paper for publication.

Author Response

The author sincerely expresses thanks for your efforts to review our paper. Also, the authors appreciate the reviewer’s insightful comments.